# Hyperspectral band selection and modeling of soil organic matter content in a forest using the Ranger algorithm

Yuanyuan Shi[1], Junyu Zhao[1], Xianchong Song[1], Zuoyu Qin[1], Lichao Wu[2], Huili Wang[1], Jian Tang[1]*

1 Guangxi Forestry Research Institute, Key Laboratory of Central South Fast-Growing Timber Cultivation of Forestry Ministry of China, Nanning, China, 2 Key Laboratory of Cultivation and Protection for Non-Wood Forest Trees of National Ministry of Education, Central South University of Forestry and Technology, Changsha, China

☯ These authors contributed equally to this work.
* lk779343445@126.com

**Data Availability Statement:** All relevant data are within the manuscript and its Supporting Information files.

## Abstract

Effective soil spectral band selection and modeling methods can improve modeling accuracy. To establish a hyperspectral prediction model of soil organic matter (SOM) content, this study investigated a forested *Eucalyptus* plantation in Huangmian Forest Farm, Guangxi, China. The Ranger and Lasso algorithms were used to screen spectral bands. Subsequently, models were established using four algorithms: partial least squares regression, random forest (RF), a support vector machine, and an artificial neural network (ANN). The optimal model was then selected. The results showed that the modeling accuracy was higher when band selection was based on the Ranger algorithm than when it was based on the Lasso algorithm. ANN modeling had the best goodness of fit, and the model established by RF had the most stable modeling results. Based on the above results, a new method is proposed in this study for band selection in the early phase of soil hyperspectral modeling. The Ranger algorithm can be applied to screen the spectral bands, and ANN or RF can then be selected to construct the prediction model based on different datasets, which is applicable to establish the prediction model of SOM content in red soil plantations. This study provides a reference for the remote sensing of soil fertility in forests of different soil types and a theoretical basis for developing portable equipment for the hyperspectral measurement of SOM content in forest habitats.

## Introduction

The measurement of soil organic matter (SOM) content is a critical procedure in carbon cycle studies and forest management. Using spectroscopic techniques instead of laboratory analysis method can significantly improve the timeliness of the SOM determination process. However, the spectral response of SOM features many narrow wave bands and a large amount of data, resulting in information redundancy. Moreover, many environmental factors can affect the spectral response of SOM, such as soil water content, soil iron content, air temperature, and

**Funding:** This work was supported by the "Guangxi Key Laboratory of Superior Timber Trees Resource Cultivation, grant number 2020-A-04-01" and " Innovation- Driven Development Special Fund Project of Guangxi, grant number AA17204087-11". Findings or opinions presented in this work are not endorsed by, nor do they represent the views of, either funding agency.

**Competing interests:** The authors have declared that no competing interests exist.

humidity [1]. These factors affect the characteristics and the analysis of the spectral response of soil, which increases the difficulty in using hyperspectral data to predict SOM content [2–5]. Related studies show that the key to improving this type of prediction accuracy lies in effective soil spectral band selection and modeling [6, 7].

In view of the above problems, a large number of studies have been carried out on spectral band selection and modeling that were designed to improve the accuracy of spectral prediction. A summary of the state-of-the-art works conducted on spectral band selection and modeling is presented in Table 1. It is evident from recent work done on soil spectral modeling that various methods have been implemented. The raw spectra may be influenced by instrument condition, measurement environment, and sample conditions. Spectral pre-processing is an important step in soil spectra analysis. Pretreatment removes extraneous interference and improves the performance of estimation models. Common spectral pre-processing techniques include mathematical transformation [8–12], Savitzky–Golay (SG) [11, 13–15], continuum removal (CR) [15–17], multiplicative scatter correction (MSC) [11, 17], and standard normal variate (SNV) [11, 17]. Spectral band selection aims to select the optimal variables from the raw spectra, in order to enhance the spectral sensitivity of soil properties. Several Machine learning (ML) methods have been proposed for spectral band selection, such as competitive adaptive reweighting sampling (CARS) [8, 16–18], principal component analysis (PCA) [14, 15, 19, 20], locally linear embedding (LLE) [14], multidimensional scaling (MDS) [14], meta-heuristic algorithms [21], and rough set algorithms [22]. In addition, partial least squares regression (PLSR) [13, 20, 23], artificial neural networks [4], random forest (RF) [2, 17, 24], and support vector machine (SVM) [24, 25] are common methods of soil spectral modeling.

After reviewing the state-of-the-art literature, it is evident that exploring the most suitable band selection and selecting the appropriate combination of modeling methods are still the key problems in the field of hyperspectral prediction modeling of soil properties. ML techniques have proved to be effective in dealing with large amounts of soil spectral variables [26–29]. The techniques mentioned above have been applied to obtain prediction models of soil properties. However, few studies have mentioned the application of Ranger and least absolute shrinkage and selection operator (Lasso) algorithms to spectral band selection. These two algorithms are suitable for the resampling and feature selection of high-dimensional data with many features [30, 31]. Moreover, these two algorithms have been applied well in disease diagnosis and image processing [32, 33]. One of the goals of this study was to determine whether ML techniques can improve the selection of spectral bands.

Many studies have shown that models should be established separately for different soil types and land use patterns [34]. The factors involved in soil formation in different regions vary. Additionally, land use patterns can result in changes in soil properties [35, 36]. Furthermore, the source and composition of SOM also affect the spectral response. Therefore, given different soil types [5, 37], vegetation types [38, 39], and land use patterns [40, 41], the methods of establishing the SOM spectral inversion model will vary. Studies have been conducted on the spectral characteristics of SOM content associated with different land use patterns, such as alpine meadows [42, 43] and coal mining areas [44]. Forest SOM is mainly formed by the decomposition of dry branches and fallen leaves, and it is one of the important indicators that can be used to measure the fertility and quality of forest soil. SOM not only contains a variety of nutrients necessary for the growth and development of trees, but also has the potential to reduce levels of pollution and heavy metals in soil [45–47]. Information about the spatial–temporal variation of SOM content provides the basis of accurate fertilization and information management. Hence, the real-time acquisition of SOM content in different types of soil and forests is of great significance to forest management. Some forestry researchers have constructed hyperspectral estimation models of forest SOM content [48, 49].

**Table 1. Summary of spectral band selection and modeling techniques for soil properties prediction.**

| Reference | Spectral pre-processing techniques | Band selection methods | Modeling methods | Predicted properties | Area of investigation/soil type/vegetation | $R^2$/RMSE of best pre-processing method |
|---|---|---|---|---|---|---|
| [13] | MSC, SNV, SG-FD SG-SD | | PLSR, RF, SVR, MARS | SOC | India, middle Indo-Gangetic plain / silty clay loam/ rice cultivation | 0.73/0.07 |
| | | | | | | PLSR |
| [15] | SG, CR | PCA, Optimal band combination algorithm | Linear models, RF | SOM | China, Junggar basin/ arenosols, solvents and gypsisols/coal Mining Area | 0.93/2.52 |
| | | | | | | Optimal band combination algorithm |
| [8] | FD, Log (1/R), MC, MSC, SNV | RF, CARS | PLSR | SOC | Jianghan Plain, China/ cropland, woodland, and meadows | 0.83/2.94 |
| | | | | | | RF+ log(1/R) |
| [14] | MSC, SG, WPT | PCA, MDS, LLE | PLSR | SOM | Heilongjiang, China/Luvisols, Phaeozems/cropland | 0.85/0.28 |
| | | | | | | SGF+PCA |
| [20] | R, SG, WPD, ND | NDR, SWDR, PCA, PCADR | PLSR | SOM | Jilin province, China/meadow soil, black soil, white soil, and paddy soil/ corn-cultivated land | 0.83/0.21 |
| | | | | | | ND-R-SWDR |
| [9] | MSC, MC, SG, MA, FD, SD, CR, Log(1/R) | | AdaBoost, RR, KRR | SOM | Northwest China/alluvial soils and irrigated soils/farmland | 0.91/0.22 |
| | | | | | | AdaBoost-KRR |
| [16] | CR, FD | CARS, GA | PLSR, Semi-DNNR | SOM | Jilin province, China/saline-alkali land/ Coal Mining Area | 0.71/3.52 |
| | | | | | | Semi-DNNR |
| [17] | R, CR, FDR | CARS | RF | SOM | Songnen Plain, China /Phaeozems, Chernozems, Arenosols, and Cambisols/ | 0.89/0.42 |
| | | | | | | FDR |
| [24] | SG, SNV, MSC, CR, NBR | | SVM, RF, ANN, WAPLS | SOC | Santa Catarina State, Brazil | 0.82/0.48 NBR +WAPLS |
| [11] | SNV, MSC, SG | | PLSR, MLP | SOC | Sygera Mountains, China/Cambisols, Luvisols, Phaeozems, and Umbrisols | 0.92/6.22 MLP |
| [25] | SG, FD, SD | | PLSR, SVMR | PTEs (Cu, Mn, Cd, Zn, Fe, Pb and As) | Bílina and Tušimice, Czech Republic/ Vertisols and Chernozems | 0.89/1.89 FD+SVMR |
| [10] | MSC, SG, SNV log10 (1/x), | | PLSR | SOC | Amazon, Brazil/the area includes: Argisols, Spodosols, Neosols, Planosols | 0.71/5.69 |
| | | | | | | SNV |
| [12] | SG, FD, SD, RL | | BPNN, PLSR, GA-BPNN | TN, TP, TK | Guangdong, China/Lateritic red soil | 0.90/25.42 |
| | | | | | | GA-BPNN |

Note: MSC (multiplicative scatter correction), SNV (standard normal variate), SG-FD (Savitzky–Golay smoothing first derivative), SG-SD (Savitzky–Golay smoothing second derivative), PLSR (partial least-squares regression),RF (random forest), SVR (support vector regression), MARS (multivariate adaptive regression splines), SOC (soil organic carbon), CR (continuum removal), PCA (principal component analysis), MC (mean centering), CARS (competitive adaptive reweighted sampling), WPT (wavelet packet transform), MDS (multidimensional scaling), LLE (locally linear embedding), WPD (wavelet packet denoising), ND (no denoising), NDR (no dimensionality reduction), MA (moving average), RR (ridge regression), KRR (kernel ridge regression), Semi-DNNR (semi-supervised deep neural network regression), GA (genetic algorithm), NBR (normalization by range), WAPLS (weighted averaging partial least squares regression), MLP (multilayer perceptron), RL (reciprocal Logarithmic), BPNN (back-Propagation neural network)

As already mentioned, currently available hyperspectral modeling methods are relatively mature, and many methods can be used to select hyperspectral bands. However, few studies have addressed the selection of hyperspectral sensitive bands using ML algorithms, such as Ranger and Lasso. In addition, few current studies have focused on hyperspectral modeling of SOM content in red soil plantations. Therefore, the motivations behind the present work included the following:

- Improving the performance of hyperspectral prediction modeling of SOM using hyperspectral band selection methods (Ranger or Lasso).

- Comparing the performance of hyperspectral models developed with PLSR, RF, SVM, and ANN in predicting the SOM content of red soil plantations.

Based on the motivations mentioned above, in this work, Ranger and Lasso algorithms were introduced to select optimal hyperspectral bands. Then, the optimal dataset with reduced features was trained using several modeling methods. The main contributions of this work were as follows:

- The ML techniques (Ranger or Lasso) were adopted to select the optimal hyperspectral bands of forest SOM and improve the modeling accuracy.

- The hyperspectral prediction models established using the RF and ANN algorithms could estimate the SOM content in red soil plantations.

- Nonlinear modeling algorithms, RF and ANN, performed better than PLSR and SVM. Particularly, RF better adapted to datasets of different sizes than the other algorithms.

## Materials and methods

### Study area

The study area is located in the Guangxi State-owned Huangmian Forest Farm, Liuzhou City, China (24˚37'–24˚52' N and 109˚43'–109˚58' E). The area is characterized by a humid subtropical monsoon climate with an annual mean rainfall of 1,750–2,056 mm and an annual mean temperature of 20.4˚C. The average relative humidity is 80%, and the annual average sunshine is 1596.8 h. The soil type of the study area is red soil (Allitic-Udic Ferrosols), which is derived from Devonian sandstones and arenaceous shale, according to the Chinese Soil Taxonomic Classification. The location map of the study area is shown in Fig 1.

### Collection and measurement of soil samples

The flowchart of this study is shown in Fig 2. A total of 104 soil samples were collected from a typical Eucalyptus plantation covering an area of 8666.67 ha. All soil samples were red soil,

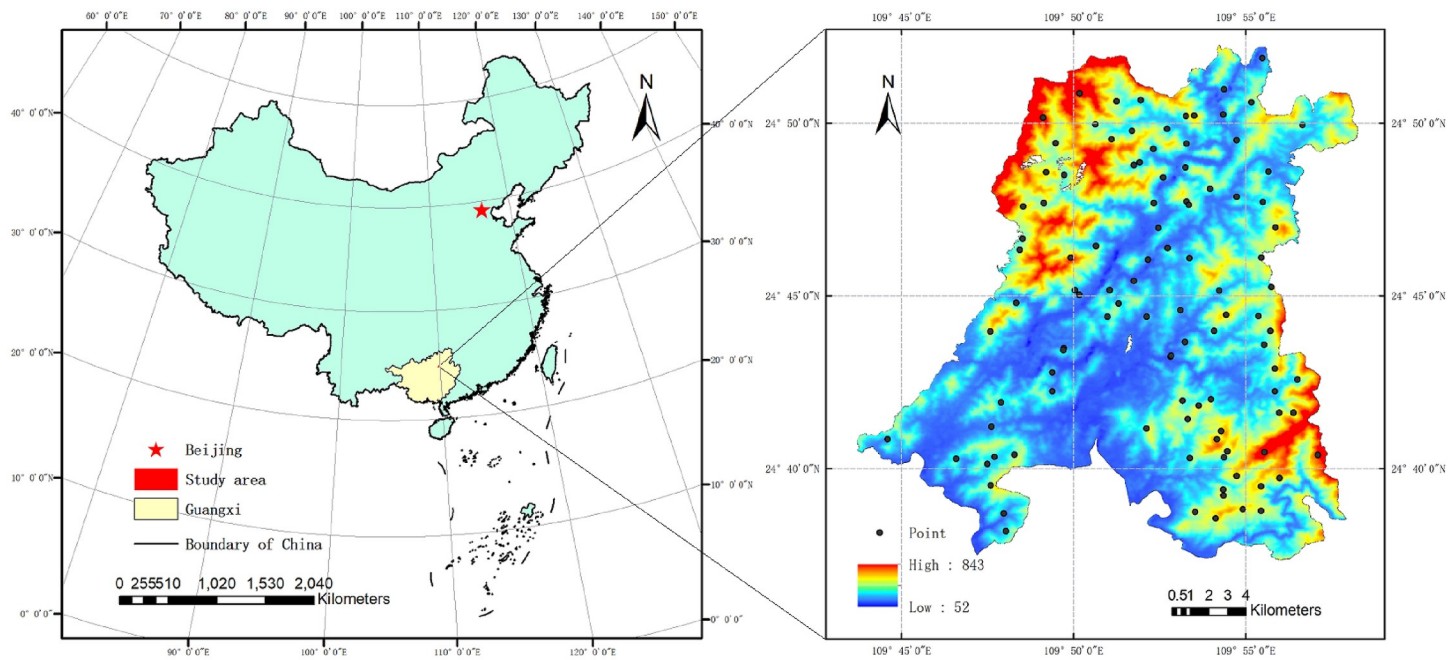

**Fig 1. Location of study area and distribution of soil sampling sites.**

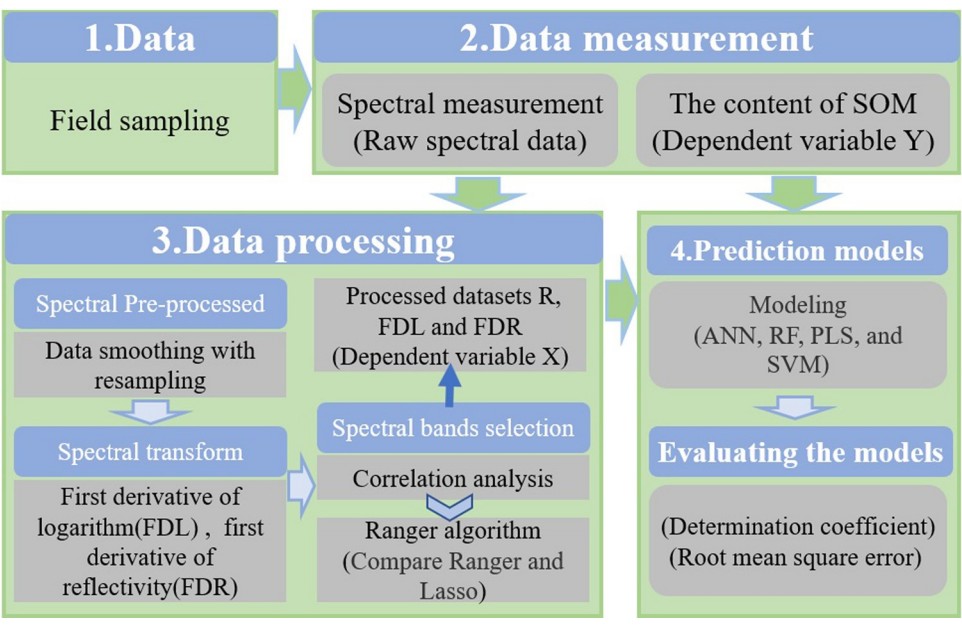

**Fig 2. The flowchart of the paper.**

with acid sedimentary rock being the soil-forming parent material, and the aboveground vegetation dominated by the Eucalyptus plantation (Fig 1). During the field sampling, samples of the 0–20 cm layer of topsoil were collected using the S-type sampling method due to the fact that the plantation had uniformly growing trees and infrequent human activities. Intruding plant parts and macroscopic animals were removed from the soil during the collection of soil samples. After mixing the soil samples, 500 g soil was collected by quartering. Each air-dried soil sample was placed into an individual self-sealing bag and then labeled. After being ground in the lab, part of the soil was sifted using a 0.2-mm soil sieve and then heated by the potassium dichromate method to measure the SOM content; the other part was sifted through a 0.149-mm soil sieve for the collection of indoor hyperspectral data. When modeling, the dataset was divided into a training set and a verification set using a ratio of 7:3. The statistics related to the characteristics of SOM are shown in Table 2.

## Collection and preprocessing of hyperspectral datasets

The soil hyperspectral data were collected by a ASD FieldSpec® 4 Hi-Res ground object spectrometer (Analytical Spectral Devices, Boulder, CO, USA). The spectral band included the visible light and near-infrared regions (350–2500 nm) with a resolution of 1 nm; the probe had a field angle of 15˚, a 50 W halogen lamp equipped with the spectrometer served as the light source, and the incident angle of the light source was 45˚. Samples were loaded into a 7 cm diameter sample cup that was about 1.5 cm deep. The soil samples were flattened and compacted. With the probe being 5 cm away from the soil surface, the soil spectral data were

**Table 2. Statistical characteristics of soil organic matter content of a *Eucalyptus* plantation in the red soil area of Huangmian Forest Farm.**

| Data | Sample | Range (g/kg) | Mean (g/kg) | SD | Skewness | Kurtosis | CV (%) |
|---|---|---|---|---|---|---|---|
| Modeling set | 76 | 4.88–48.96 | 30.46 | 7.31 | 0.247 | −0.71 | 23.99 |
| Validation set | 28 | 13.94–52.88 | 32.21 | 2.13 | 0.365 | −0.928 | 6.61 |

measured after the air background value was deducted by the instrument controller. To ensure the accuracy of the data, collection of the spectral data of each soil sample was repeated ten times; the average value was taken as the final soil spectral data. ViewSpecPro 6.0 spectral data processing software was used to preprocess the spectral data. First, the bands at the two ends (<400 nm and >2400 nm) of the measuring range of the spectrometer where the spectral data were noisy were removed because of the poor performance of the spectrometer in this range of the spectrum; then, the first derivative of logarithm (FDL) and first derivative of reflectivity (FDR) were calculated based on the raw spectral reflectance (R) of the soil.

## Hyperspectral band selection

The present study involved screening the hyperspectral bands twice before the establishment of prediction models. First, the correlations between the SOM content and R, FDR, and FDL were analyzed separately, and a significance test was performed using $p = 0.01$ to screen and select the significant bands. Second, the Ranger and Lasso algorithms were used to further select the sensitive bands. Ranger [30], as a factor selection method based on the RF algorithm, calculates the importance of each band and retains the bands that are effective in identifying the desired information as the set of independent variables for modeling. In this study, with an importance value of 30 as the baseline, each band with an importance value of greater than 30 was reserved as a sensitive band. As a data dimensionality reduction method, Lasso [50] is suitable for both linear and nonlinear cases. Lasso selects variables from the sample data based on the penalty method. By compressing the original coefficients, the smaller coefficients are compressed to 0; therefore, the variables corresponding to these coefficients are considered as insignificant variables and are eliminated. The sensitive bands selected this way by Lasso served as the independent variables for modeling, and were combined with the SOM content to constitute the modeling dataset for modeling analysis. The Ranger and Lasso algorithms were implemented using RStudio software, and correlation analysis was performed in SPSS19.0 (SPSS Inc., Chicago, IL, USA).

## Modeling methods

In this study, with the SOM content as the dependent variable and the spectral data of the sensitive bands as the independent variable, PLSR, RF, an SVM, and an ANN were used to establish the models. The accuracy of the models was measured by the determination coefficient ($R^2$) and root mean square error (RMSE). The smaller the RMSE, the stronger the estimation ability of the model; the closer $R^2$ was to 1, the better the stability and the higher the goodness of fit of the model. When modeling, the dataset was divided into a training dataset and a verification dataset using a ratio of 7:3. Each algorithm validated the models by means of 5-fold cross validation. All four modeling algorithms used in this study were implemented using Rstudio software.

**Partial least squares regression.** With the goal of using regression to model the data relationships that involve multiple dependent and independent variables, PLSR is applicable to the processing of abundant sample data [20, 51]. Moreover, when combined with the advantages of PCA, PLSR can eliminate some of the variation in the data and extract the information in a way that is most effective in analyzing the data. In addition, PLSR has been extensively used in spectral analysis modeling to establish a linear regression model between predictive and observation variables.

**Artificial neural network.** An ANN, a model composed of multiple nonlinear elements (neurons), is composed of multi-layer structures that are further connected to form a huge

network of connections [52]. The feed forward-back propagation neural network used in this study was an ANN model suitable for the general network.

**Support vector machine.** An SVM, as an efficient and superior type of algorithm, can avoid the overfitting and local optimization of empirical nonlinear methods. With a strong ability to generalize, SVMs have been widely used in prediction efforts in various fields. Their greatest strength is that they only require a small number of samples to yield excellent results [53]. The key to using an SVM for regression prediction lies in parameter optimization. In this study, the R language was applied to realize the automatic optimization of the SVM parameters, thereby obtaining the optimal SVM prediction model.

**Random forest.** When monitoring land surface information using remote sensing, RF is an algorithm that can quantitatively analyze ground-based information. RF, which integrates the advantages of a decision tree and bagging set, introduces randomly selected features into model training [54]. Using RF to train the models not only effectively avoids overfitting, but also restrains the negative effects of noise. This improves the accuracy and stability of model classification and prediction. In addition, RF is able to process and analyze high-dimensional information features, and statistically analyze the importance of multi-dimensional features, which is conducive to the comprehensive use of the hyperspectral characteristics of ground-based objects. Since the number of decision trees has a great influence on a model, the number of decision trees was set as 100 in this study to guarantee the stability of each model.

## Results

### Hyperspectral characteristics of forest soil organic matter

Based on the analysis of the hyperspectral curve and the mathematical transformation of soil data in the red soil plantation, a negative correlation was observed between spectral reflectance and SOM content. From the entire range of the bands, the curves of the SOM content and of R shared a largely consistent trend. The R decreased significantly with an increase in SOM content, indicating a negative correlation between the two (Fig 3A). The slope of the curve in the visible light band (400–780 nm) was relatively large, while the curve varied slightly in the near-infrared band (780–2500 nm). The spectral curve had absorption peaks in the bands of 1300–1400 nm, 1750–1850 nm, and 2250 nm. As for the FDL and FDR spectral curves, the distances between the spectral curves of soil with different levels of SOM content decreased, and the number of characteristic absorption peaks increased, with the absorption peaks becoming sharper. However, the curves crossed in the region between the visible light and shortwave infrared bands (Fig 3B and 3C). The noise of FDL and FDR spectral curves was large in the

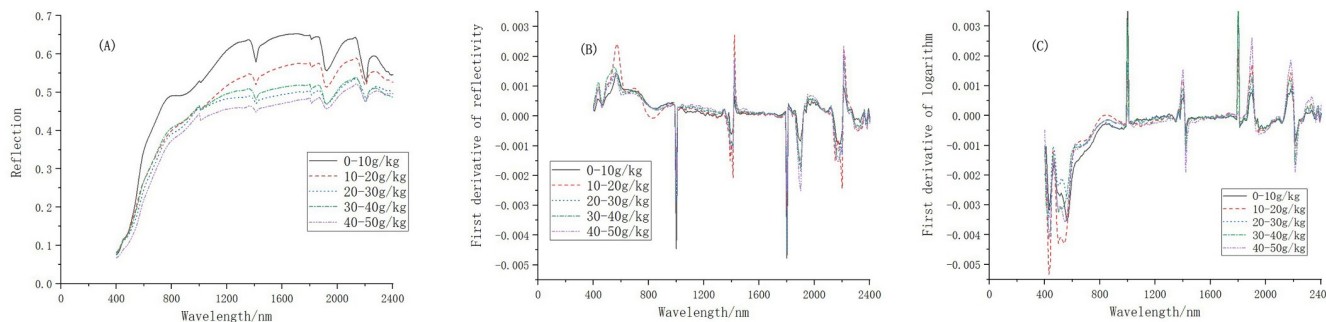

**Fig 3. Average spectral curve of:** (A) soil reflectance; (B) the first derivative; and (C) the logarithmic first derivative with different levels of organic matter content.

visible light (400–650 nm) and near the 2400 nm bands, and new absorption peaks were found in the 1000 nm band.

## Correlation between spectral reflectance and forest SOM

The analysis of the correlations between the SOM content and R, FDR, and FDL showed that the mathematical transformation improved the band sensitivity differently from the use of raw spectral reflectance (Fig 4). In addition, R was negatively correlated with the SOM content. A significantly negative correlation was observed between R and SOM content in bands of 590–1200 nm, although the correlation between these two in the other bands was not significant. FDL improved the hyperspectral sensitivity of the 750–800 nm and 1200–1600 nm near-infrared bands, whereas FDR enhanced the sensitivity of the visible light and mid-infrared bands. Through correlation analysis, the bands where the SOM content was significantly correlated with R (600–1150 nm), FDR (500–540 nm and 630–680 nm), and FDL (980–1000 nm, 1250–1350 nm, and 1550 nm) were selected as the candidate bands to be used for preliminary screening for hyperspectral selection in the next step. The R, FDR, and FDL values corresponding to these candidate bands formed the three preliminary modeling datasets.

## Selection of forest soil organic matter sensitive to hyperspectral bands

The ML algorithms Ranger and Lasso were used to screen the spectral bands of the preliminary datasets. The results showed that the modeling accuracy could be improved if Ranger was applied to screen the spectral bands. Based on the R, FDR, and FDL preliminary datasets screened by correlation analysis, Ranger and Lasso were used to perform dimensionality

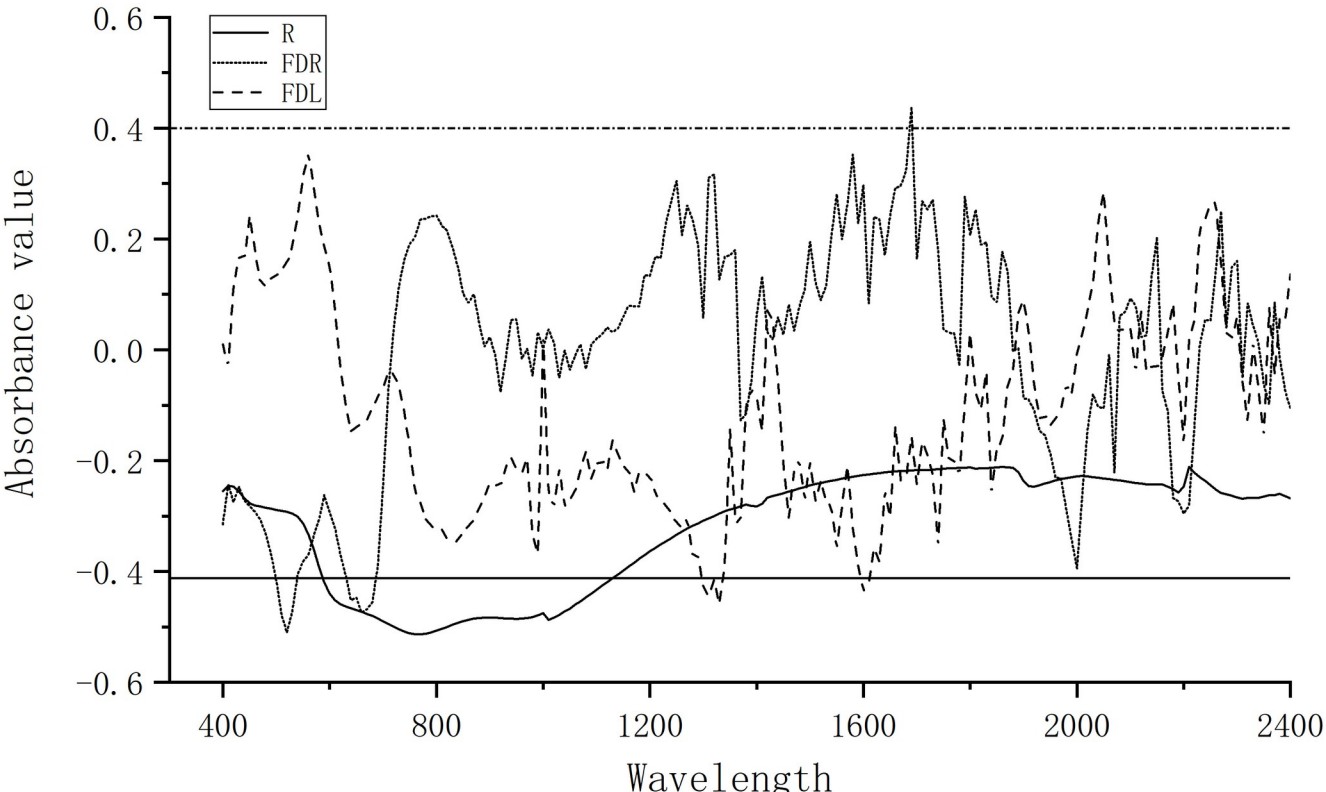

**Fig 4. Correlation coefficient between soil organic matter and spectral reflectance.**

reduction on spectral bands corresponding to the three preliminary datasets, and the selected spectral bands formed three datasets. Using RF-based modeling, the strengths and weaknesses of Ranger and Lasso in selecting the hyperspectral bands were compared.

The results show the modeling of the two datasets filtered by the Ranger and Lasso algorithms (Table 3). Both Ranger and Lasso significantly increased the $R^2$ values of modeling and improved the modeling accuracy. The dataset obtained by Ranger had a training $R^2$ greater than that of the dataset obtained by Lasso. A slight difference in the training $R^2$ was observed between the FDR datasets obtained by Ranger and Lasso. However, the R and FDL screened by Ranger outperformed those screened by Lasso in terms of the training $R^2$. Therefore, in this study, Ranger was applied to filter the hyperspectral bands of the R, FDL, and FDR datasets to obtain the optimal hyperspectral sensitive bands of forest SOM, thereby obtaining the final datasets for modeling.

## Establishment of a hyperspectral prediction model of forest SOM

Based on the modeling results, an ANN and RF achieved satisfactory results in modeling, the model jointly established by FDR and ANN yielded the optimal results, and the RF modeling method produced satisfactory results for the three final datasets (Table 4). The PLSR modeling results showed a training $R^2$ of more than 0.95 while the verification $R^2$ was relatively small and the RMSE value was large, indicating overfitting. The training results of the other three algorithms were better than the verification results. Among the SVM modeling results, the R and FDR datasets featured a comparatively low training $R^2$, while the FDL dataset had a training $R^2$ of 0.82, which was greater than the $R^2$ values of the other two datasets. As for the ANN modeling results, the training $R^2$ of the FDL dataset was relatively low, while the FDR dataset had a training $R^2$ of 0.92. According to the RF modeling results, all three datasets featured a training $R^2$ of above 0.85, a verification $R^2$ of above 0.7, and an RMSE of less than 5, which was smaller than the RMSE fitted by the SVM and ANN algorithms. Based on the model fitting results of the R, FDR, and FDL datasets, the model constructed by ANN combined with the FDR dataset had a high prediction accuracy, and the models established by RF combined with the other two datasets featured a relatively high level of accuracy. In general, the model established by an ANN algorithm combined with the FDR dataset had the highest prediction accuracy, the model constructed by RF combined with the R and FDL datasets featured the second highest prediction accuracy, and the model established by RF was comprehensively stable.

## Discussion

### Methods of selecting hyperspectral bands

The results showed that the spectral band selection method based on the combination of an ML algorithm, i.e. Ranger or Lasso, and mathematical transformation remarkably improved the modeling accuracy when using spectral data., A previous study [2] showed that differential transformation contributed to the decomposition of sensitive information from the soil spectrum, thereby effectively improving the accuracy of spectral analysis. The differential

**Table 3. Comparison of modeling results between two filter algorithms.**

| Filtering algorithm | R | | | FDL | | | FDR | | |
|---|---|---|---|---|---|---|---|---|---|
| | Training $R^2$ | Validation $R^2$ | RMSE | Training $R^2$ | Validation $R^2$ | RMSE | Training $R^2$ | Validation $R^2$ | RMSE |
| No filtering | 0.59 | 0.43 | 8.35 | 0.61 | 0.48 | 7.56 | 0.64 | 0.50 | 8.65 |
| Ranger | 0.88 | 0.78 | 4.97 | 0.85 | 0.73 | 4.63 | 0.89 | 0.75 | 4.98 |
| Lasso | 0.85 | 0.73 | 5.88 | 0.84 | 0.68 | 5.31 | 0.86 | 0.74 | 4.95 |

**Table 4. Comparison of soil organic matter hyperspectral prediction modeling methods.**

| Differential transform | PLSR | | | RF | | | SVM | | | ANN | | |
|---|---|---|---|---|---|---|---|---|---|---|---|---|
| | Training $R^2$ | Validation $R^2$ | RMSE | Training $R^2$ | Validation $R^2$ | RMSE | Training $R^2$ | Validation $R^2$ | RMSE | Training $R^2$ | Validation $R^2$ | RMSE |
| R | 0.97 | 0.16 | 49.79 | 0.88 | 0.78 | 4.97 | 0.57 | 0.39 | 6.94 | 0.78 | 0.67 | 6.05 |
| FDR | 0.95 | 0.08 | 11.46 | 0.85 | 0.73 | 4.63 | 0.54 | 0.38 | 7.21 | 0.92 | 0.67 | 5.21 |
| FDL | 0.97 | 0.09 | 28.11 | 0.89 | 0.75 | 4.98 | 0.82 | 0.64 | 7.68 | 0.51 | 0.46 | 7.51 |

transformation of the spectrum weakened the influence of multiplicative factors caused by the variation in light conditions [39]. In addition, FDR processing eliminated the interference of background noise, decomposed the mixed overlapping peaks, and improved the spectral resolution and sensitivity, making it easy to locate the highly correlated bands [55]. In this study, the correlation coefficient obtained after differential transformation also indicated that mathematical transformation enhanced the sensitivity of some spectral bands, which was consistent with the conclusions in previous studies. Like mathematical transformation, the selection of characteristic bands is one of the approaches that can be used to improve the accuracy of spectral analysis. Many methods can be used to select the characteristic bands, such as PCA and the correlation coefficient method, the purpose of which is to select the bands sensitive to SOM so as to improve the prediction accuracy of the model.

In addition, the ML algorithms were used for the selection of hyperspectral bands in this paper. The most common tasks in ML research are regression [56], classification [57–59], clustering [60], and dimensionality reduction [61, 62]. ML algorithms are widely used in research fields including plant diseases [63], disease diagnosis [57], and remote sensing image processing [64–66]. The present study proposed a hyperspectral band selection method realized by the ML algorithms (Ranger and Lasso) based on traditional mathematical transformation. This method can overcome the shortcomings of hyperspectral data to some extent, such as information redundancy caused by having a large amount of information, and improve the accuracy of spectral modeling. This can provide a new approach for hyperspectral band selection during the early phase of soil hyperspectral modeling.

Based on the results obtained in the present study, both algorithms improved the accuracy of hyperspectral modeling to varying degrees. Through a comparison of the algorithms, it was found that, considering the datasets used in this study, Ranger combined with mathematical transformation was the optimal method of selecting the spectral bands, which effectively improved the accuracy of hyperspectral prediction of the SOM content in a red soil plantation.

## Comparison of linear and non-linear modeling algorithms

Hyperspectral prediction models established using RF and ANN algorithms could estimate the SOM content in red soil plantations. In addition, PLSR, RF, SVM, and ANN are the four modeling methods shown to have excellent performance in studies on the establishment of hyperspectral prediction models [20, 51, 67, 68], among which PLSR is the most extensively applied linear fitting method [29]. However, the relationship between the SOM content in an area and related spectral features should be more complex than a simple linear relationship. Hence, RF, an SVM, and an ANN, algorithms that have been widely used in recent years, were compared in the present study. Although the model established by PLSR presented a phenomenon of overfitting, it was unjustifiable to conclude that this method was not suitable for spectral modeling. The overfitting phenomenon might have been caused by the low amount of data available [69, 70]. If conditions permit, increasing the size of the training dataset may prevent the overfitting phenomenon. Nevertheless, attention should also be paid to such phenomena, because applying

PLSR to establish a model with a small amount of data can lead to overfitting. In terms of the modeling results, the nonlinear modeling algorithms, RF and ANN, performed better, especially RF, which was more tolerant of data preprocessing and the amount of modeling data available. In this study, RF achieved the best modeling results based on an analysis the three datasets employed here. It can be inferred that the linear fitting method requires a larger amount of data, while the nonlinear modeling algorithm is more tolerant to a small amount of data.

## Conclusions

First, the results obtained in this study showed that the hyperspectral band selection method featuring the integration of mathematical transformation and ML (Ranger and Lasso) effectively improved the accuracy of hyperspectral modeling and overcame the shortcomings of hyperspectral data, such as information redundancy caused by having a large amount of information. After the FDR transformation, the sensitivity of the 750–800 nm and 1200–1600 nm near-infrared bands was improved. The mathematical transformation of FDL enhanced the sensitivity of the visible light and the mid-infrared bands. Therefore, mathematical transformation expanded the range of spectral reflectance sensitive bands and improved the modeling accuracy. By further selecting bands using either Ranger or Lasso, the accuracy of modeling could be significantly improved. Second, the comparison of several modeling methods showed that the spectral modeling technique was applicable to predicting the SOM content in red soil plantations. When PLSR was used to establish the models based on the R, FDR, and FDL datasets, overfitting occurred. The modeling results of an SVM were not as ideal as those of RF and an ANN. By comparing the modeling results of RF and the ANN, it was found that the optimal model was established by the ANN combined with the FDR dataset. However, it is worth mentioning that RF realized stable modeling results when combined with any of the three datasets. When the model was established based on the R and FDL datasets, RF achieved the optimal modeling results. However, when the FDR dataset was used for modeling, the ANN realized the best modeling results. Based on the results obtained in this study, when studying the hyperspectral prediction of the SOM content in red soil plantations with a small sample size, mathematical transformation should be combined with the Ranger algorithm to screen the hyperspectral data, and nonlinear algorithms, such as RF and ANN, should be applied for modeling.

The present study results provide a method for rapidly predicting the SOM content in forest soil. In addition, hyperspectral band selection greatly reduced the number of bands involved in the calculation and improved the efficiency and accuracy of spectral modeling. Considering that soil samples from different soil types and land use patterns usually exhibit significant spectral differences, it may not be feasible to extrapolate our results to other study areas. As future work, the proposed methods could be implemented on hyperspectral image band selection and modeling. This technique is expected to provide technical support for the non-destructive and large-scale monitoring of forest soil health assessment.

## Supporting information

**S1 Datasets.**
(XLSX)

## Acknowledgments

We thank each editor and the anonymous reviewers for their insightful comments, which helped in the publication of this manuscript.

## Author Contributions

**Data curation:** Yuanyuan Shi, Junyu Zhao.

**Formal analysis:** Yuanyuan Shi, Junyu Zhao, Xianchong Song, Zuoyu Qin.

**Investigation:** Yuanyuan Shi, Junyu Zhao, Xianchong Song, Zuoyu Qin, Huili Wang, Jian Tang.

**Resources:** Junyu Zhao.

**Software:** Yuanyuan Shi, Junyu Zhao.

**Supervision:** Lichao Wu, Jian Tang.

**Writing – original draft:** Yuanyuan Shi.

**Writing – review & editing:** Yuanyuan Shi.

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
