## [Decision Letter · Decision Letter 0]

12 Apr 2021

PONE-D-21-09521

Hyperspectral Band Selection and Modeling of Soil Organic Matter Content in a Forest Using the Ranger Algorithm

PLOS ONE

Dear Dr. Tang,

Thank you for submitting your manuscript to PLOS ONE. After careful consideration, we feel that it has merit but does not fully meet PLOS ONE’s publication criteria as it currently stands. Therefore, we invite you to submit a revised version of the manuscript that addresses the points raised during the review process.

Based on the reviewer's and my own suggestions, I recommend major revisions for this paper.

We look forward to receiving your revised manuscript.

Kind regards,

Thippa Reddy Gadekallu

Academic Editor

PLOS ONE

Journal Requirements:

2. We note that Figure 1 in your submission contain map images which may be copyrighted. All PLOS content is published under the Creative Commons Attribution License (CC BY 4.0), which means that the manuscript, images, and Supporting Information files will be freely available online, and any third party is permitted to access, download, copy, distribute, and use these materials in any way, even commercially, with proper attribution. For these reasons, we cannot publish previously copyrighted maps or satellite images created using proprietary data, such as Google software (Google Maps, Street View, and Earth). For more information, see our copyright guidelines: http://journals.plos.org/plosone/s/licenses-and-copyright.

2.1.    You may seek permission from the original copyright holder of Figure 1 to publish the content specifically under the CC BY 4.0 license. 

2.2.    If you are unable to obtain permission from the original copyright holder to publish these figures under the CC BY 4.0 license or if the copyright holder’s requirements are incompatible with the CC BY 4.0 license, please either i) remove the figure or ii) supply a replacement figure that complies with the CC BY 4.0 license. Please check copyright information on all replacement figures and update the figure caption with source information. If applicable, please specify in the figure caption text when a figure is similar but not identical to the original image and is therefore for illustrative purposes only.

Reviewers' comments:

Reviewer's Responses to Questions

**Comments to the Author**

1. Is the manuscript technically sound, and do the data support the conclusions?

Reviewer #1: Yes

Reviewer #2: Yes

2. Has the statistical analysis been performed appropriately and rigorously? 

Reviewer #1: Yes

Reviewer #2: Yes

3. Have the authors made all data underlying the findings in their manuscript fully available?

Reviewer #1: Yes

Reviewer #2: Yes

4. Is the manuscript presented in an intelligible fashion and written in standard English?

Reviewer #1: Yes

Reviewer #2: Yes

5. Review Comments to the Author

Reviewer #1: "The main aim of this study is to explore and highlight the role of AI in Civil Engineering and Construction Sector. my commnets are

1.There are many grammatical mistakes and spelling mistakes in this paper. please try to improve it.

2.On many occasions, somewhat unscientific statements are made that are somewhat lacking in substance and that make the work unworthy. Excessive basic materials that do not contribute anything from a scientific point of view. justyfy it/explain it.

3. An easy-to-understand flow chart is important.

4. Some of the assumptions made in the paper are not clear. They need a better motivation and justification.

5. Text and formulas present typos that need to be corrected.

6. Recent references should be cited (last 1 year).

7. Authors should highlight the limitations of the proposed approach.

8. Certain sections of the text describe information that is out of context or does not appear to be appropriate for that section of the text. "SPLIT THE CONCLUSION INTO DISCUSSION AS WELL."

9. some related refrences need to be cited to borden the scope of paper.

H. Xu, H. Yang, Q. Shen, J. Yang and H. Liang, “Automatic terrain debris recognition network based on 3d remote sensing data,” Computers, Materials & Continua, vol. 65, no. 1, pp. 579–

596, 2020.

Sharma, Sparsh, Suhaib Ahmed, Mohd Naseem, Waleed S. Alnumay, Saurabh Singh, and Gi Hwan Cho. "A Survey on Applications of Artificial Intelligence for Pre-Parametric Project Cost and Soil Shear-Strength Estimation in Construction and Geotechnical Engineering." Sensors 21, no. 2 (2021): 463.

X. Liu, J. Yu, W. Song, X. Zhao, L. Zhao et al., “Remote sensing image classification algorithm based on texture feature and extreme learning machine,” Computers, Materials & Continua, vol. 65, no. 2, pp. 1385–1395, 2020.

Q. He, S. Yu, H. Xu, J. Liu, D. Huang et al., “A weighted threshold secret sharing scheme for remote sensing images based on chinese remainder theorem,” Computers, Materials & Continua, vol. 58, no. 2, pp. 349–361, 2019.

Gadekallu, Thippa Reddy, Neelu Khare, Sweta Bhattacharya, Saurabh Singh, Praveen Kumar Reddy Maddikunta, In-Ho Ra, and Mamoun Alazab. "Early detection of diabetic retinopathy using PCA-firefly based deep learning model." Electronics 9, no. 2 (2020): 274.

D. Corral-Plaza, J. Boubeta-Puig, G. Ortiz and A. Garcia-de-Prado, “An internet of things platform for air station remote sensing and smart monitoring,” Computer Systems Science and Engineering, vol. 35, no.1, pp. 5–12, 2020.

Gadekallu, Thippa Reddy, Neelu Khare, Sweta Bhattacharya, Saurabh Singh, Praveen Kumar Reddy Maddikunta, and Gautam Srivastava. "Deep neural networks to predict diabetic retinopathy." Journal Of Ambient Intelligence and Humanized Computing (2020): 1-14.

Reviewer #2: 1.List out the main contributions of the current work.

2. Summarize the findings from recent works in the form of a table.

3. Some of the recent works on image processing such as the following can be discussed in the paper: "Image-Based malware classification using ensemble of CNN architectures (IMCEC), A novel PCA–whale optimization-based deep neural network model for classification of tomato plant diseases using GPU, Deep learning and medical image processing for coronavirus (COVID-19) pandemic: A survey".

4. Justify why the authors choose the ML algorithms used in the paper.

5. Compare the current work with recent state-of-the-art.

6. Discuss about the limitations and future scope of the current work.

6. PLOS authors have the option to publish the peer review history of their article (what does this mean?). If published, this will include your full peer review and any attached files.

Reviewer #1: No

Reviewer #2: No

---

## [Author Response · Author response to Decision Letter 0]

27 May 2021

Dear Editor and reviewers,

Thank you for your letter and comments about our manuscript entitled “Hyperspectral Band Selection and Modeling of Soil Organic Matter Content in a Forest Using the Ranger Algorithm” (PONE-D-21-09521). Those comments are invaluable and very helpful for revising and improving our paper. We have studied comments carefully and have made correction according to your suggestions. Response to the journal requirements:

1.Please ensure that your manuscript meets PLOS ONE's style requirements, including those for file naming.

Response: Thank you for your comments. We have checked and revised the manuscript style carefully according to the PLOS ONE’s style, including file naming. 

2. We note that Figure 1 in your submission contain map images which may be copyrighted. All PLOS content is published under the Creative Commons Attribution License (CC BY 4.0), which means that the manuscript, images, and Supporting Information files will be freely available online, and any third party is permitted to access, download, copy, distribute, and use these materials in any way, even commercially, with proper attribution. For these reasons, we cannot publish previously copyrighted maps or satellite images created using proprietary data, such as Google software (Google Maps, Street View, and Earth).

Response: Thank you for your comments. There is no copyright issue in Figure 1 of the manuscript, as it was drawn by the first author. Added, DEM data involved in the figure as a free download from NASA's Earth Observatory (public domain) : http://earthobservatory.nasa.gov/, basic geographic data derived from resources and environment science and data center of Chinese Academy of Sciences: https://www.resdc.cn/. 

The main corrections in the paper and the responds to the reviewer’s comments are as follows:

Response to the reviewer’ s comments:

Ref：PONE-D-21-09521

Title：Hyperspectral Band Selection and Modeling of Soil Organic Matter Content in a Forest Using the Ranger Algorithm

Journal: PLOS ONE

Reviewer #1: 

Comments to the Author

1.There are many grammatical mistakes and spelling mistakes in this paper. please try to improve it.

Response: Thank you for your comments. We have revised the whole manuscript carefully and tried to avoid any grammar mistakes or spelling mistakes. In addition, Our paper professionally edited for English language by LetPub Editing Services (http://www.letpub.com.cn/). The certificate of English Proofreading is proved in the appendix of the cover letter. We believe that the language is now acceptable for the review process.

2.On many occasions, somewhat unscientific statements are made that are somewhat lacking in substance and that make the work unworthy. Excessive basic materials that do not contribute anything from a scientific point of view. justyfy it/explain it.

Response: Thank you for your comments. According to your suggestion, we have checked and revised the manuscript style carefully. Some sentences with repeated or unscientific have been deleted. The deleted sentences are as follows:

a method used to distinguish different soil types based on their spectral characteristics. (Line49-50 Page 3 in the original manuscript)

because soil types are diverse and complex across the landscape. Therefore, the hyperspectral characteristics of different types of soils differ greatly from the results predicted by modeling. (Line51-52 Page 3 in the original manuscript)

as the comprehensive expression of natural environmental conditions and human activities. (Line53-54 Page 3 in the original manuscript)

Because vegetation litter provides an important source of SOM, differences in organic matter content exist among soils covered by different types of vegetation. Both vegetation types and land use patterns affect the variation of SOM content. Studies have shown that soil spectral reflectance varies with vegetation type and land use pattern. (Line54-58 Page 3 in the original manuscript)

A SOM spectral inversion model should be constructed based on difference in land use patterns and soil types. (Line60-61 Page5 in the original manuscript)

an extreme temperature of 38℃,an extreme minimum temperature of -2℃.( Line100-101 Page6 in the original manuscript)

 The hyperspectral data consist of a great number of narrow bands allowing the data to contain a massive amount of data. Hyperspectral data include a large amount of information, yet information redundancy inevitably creates a problem. (Line136-138 Page 7 in the original manuscript)

3. An easy-to-understand flow chart is important.

Response: Thank you for your comments. According to your suggestion, we add the flowchart (fig 2

)in the section 2. The figure are as follows:

Fig.2 The flowchart of the paper

4. Some of the assumptions made in the paper are not clear. They need a better motivation and justification.

Response: Thank you for your comments. According to your suggestion, we delete the hypothesis and add the motivation of the present work, and the added sentences are as follows:

The motivation behind the present work include:

Improving the performance of hyperspectral prediction modeling of SOM using hyperspectral 

band selection methods (Ranger or Lasso).

Comparing the performance of hyperspectral models developed with PLSR, RF, SVM, and ANN in predicting the SOM content of red soil plantations. (Line98-101 Page 8 in the revised manuscript)

5. Text and formulas present typos that need to be corrected.

Response: Thank you for your comments. We have revised the whole manuscript carefully and tried to avoid any present typos. We modify "PLS Regression" to "PLSR", "Organic Matter" to "SOM" and “RNN” to “ANN” and so on.

6. Recent references should be cited (last 1 year).

Response: Thank you for your comments. According to your suggestion, we add about 30 references (last 1 year) to the manuscript. The reference content is added correspondingly in section of Introduction and Discussion. At the same time, we add a table (table 1) in the Introduction, named “Summary of spectral band selection and modelling techniques for soil properties prediction”. (Line 61-68 Page5-6 in the revised manuscript)

7. Authors should highlight the limitations of the proposed approach.

Response: Thank you for your comments. According to your suggestion, we highlight the limitations of the approach in section of conclusion. The added sentences are as follows:

Considering that soil samples from different soil types and land use patterns usually exhibit significant spectral differences, it may not be feasible to extrapolate our results to other study areas. (Line363-365 Page20 in the revised manuscript)

8. Certain sections of the text describe information that is out of context or does not appear to be appropriate for that section of the text. "SPLIT THE CONCLUSION INTO DISCUSSION AS WELL."

Response: Thank you for your comments. According to your suggestion, we revise the whole manuscript carefully and try to avoid any contextual relation problem. In view of the confusion between discussion and conclusion, we move the discussion content in the conclusion of the original manuscript to the section of discussion. The moved sentences are as follows:

The present study proposed a hyperspectral band selection method realized by the ML algorithms (Ranger and Lasso) based on traditional mathematical transformation. This method can overcome the shortcoming of hyperspectral data to some extent, such as information redundancy caused by having a large amount of information, and improved the accuracy of spectral modeling. This can provide a new approach for hyperspectral band selection during the early phase of soil hyperspectral modeling. (Line306-312 Page17-18 in the revised manuscript)

9. some related refrences need to be cited to borden the scope of paper.

Response: Thank you for your comments. According to your suggestion, we add references about the application of machine learning algorithms in the introduction and discussion, including the fields of image classification, disease diagnosis, plant disease diagnosis, remote sensing image processing and so on. The added sentences are as follows:

The most common tasks in ML research are regression [56], classification [57-59], clustering [60], and dimensionality reduction [61, 62]. ML algorithms are widely used in research fields including plant diseases [63], disease diagnosis [57], and remote sensing image processing [64-66]. (Line304-306 Page 17 in the revised manuscript)

56. Sharma S, Ahmed S, Naseem M, Alnumay WS, Cho GH. A Survey on Applications of Artificial Intelligence for Pre-Parametric Project Cost and Soil Shear-Strength Estimation in Construction and Geotechnical Engineering. Sensors. 2021;21(2):463. https://doi.org/10.3390/s21020463. PMID: 33440731.

57. Sb A, Pkrm A, Qvp B, Trg A, Srks A, Clc A, et al. Deep learning and medical image processing for coronavirus (COVID-19) pandemic: A survey. Sustainable Cities and Society. 2020. https://doi.org/10.1016/j.scs.2020. 102589. PMID: 33169099.

58. Vasan D, Alazab M, an SWs, Safaei B, Zheng Q. Image-Based malware classification using ensemble of CNN architectures (IMCEC). Computers & Security. 2020;92:101748. https://doi.org/10.1016/j.cose.2020.101748.

59. He Q, Yu S, Xu H, Liu J, Du Y. A Weighted Threshold Secret Sharing Scheme for Remote Sensing Images Based on Chinese Remainder Theorem. Computers, Materials and Continua. 2019;58(2):349-61. https://doi.org/10.32604 /cmc. 2019.03703.

60. Moayedi Y, Somerset E, Fan S, Doumouras B, Teuteberg JJ. Predicting Cardiac Allograft Vasculopathy Profiles Using Machine Learning Clustering. The Journal of Heart and Lung Transplantation. 2021;40(4):S40. https://doi.org/10.1016/j.healun.2021.01.1836.

61. Gadekallu TR, Khare N, Bhattacharya S, Singh S, Alazab M. Early Detection of Diabetic Retinopathy Using PCA-Firefly Based Deep Learning Model. Electronics. 2020;9(2):274. https://doi.org/10.3390/electronics9020274.

62. Gadekallu TR, Khare N, Bhattacharya S, Singh S, Srivastava G. Deep neural networks to predict diabetic retinopathy. Journal of Ambient Intelligence and Humanized Computing. 2020;(13). https://doi.org/10.1007/ s12652-020-01963-7. 

63. Gadekallu TRR, Dharmendra Singh, Reddy M. Praveen Kumar, Lakshmanna Kuruva, Bhattacharya Sweta, Singh Saurabh, et al. A novel PCA-whale optimization-based deep neural network model for classification of tomato plant diseases using GPU. Journal of Real-Time Image Processing. Journal of Real-Time Image Processing. 2020. https://doi.org/10.1007/s11554-020-00987-8 .

64. Corral-Plaza D, Boubeta-Puig J, Ortiz G, Garcia-De-Prado A. An Internet of Things Platform for Air Station Remote Sensing and Smart Monitoring. Computer Systems Science and Engineering. 2020;35(1):5-12. https://doi.org/10.32604/csse.2020.35.005.

65. Liu X, Yu J, Song W, Zhao X, Wang A. Remote Sensing Image Classification Algorithm Based on Texture Feature and Extreme Learning Machine. Computers, Materials and Continua. 2020;65(2):1385-95. https://doi.org/10.32604/ cmc.2020.011308.

66. Xuhan, Yang H, Shen Q, Yang J, Shuang C. Automatic Terrain Debris Recognition Network Based on 3D Remote Sensing Data. Computers, Materials and Continua. 2020;65(1):579-96. https://doi.org/10.32604/cmc.2020. 011262.

Reviewer #2: 

Comments to the Author

1. List out the main contributions of the current work.

Response: Thank you for your comments. According to your suggestion, we add the main contributions of the current work, and the added sentences are as follows:

The main contributions of this work were as follows:

The ML techniques (Ranger or Lasso) were adopted to select the optimal hyperspectral bands of forest SOM and improve the modeling accuracy.

The hyperspectral prediction models established using the RF and ANN algorithms could estimate the SOM content in red soil plantations.

Nonlinear modeling algorithms, RF and ANN, performed better than PLSR and SVM. Particularly, RF better adapted to datasets of different sizes than the other algorithms. 

(Line105-110 Page 6 in the revised manuscript)

2. Summarize the findings from recent works in the form of a table.

Response: Thank you for your comments. According to your suggestion, we add a table (table 1) as “Summary of spectral band selection and modelling techniques for soil properties prediction”. (Line 61-68 Page5-6 in the revised manuscript)

3. Some of the recent works on image processing such as the following can be discussed in the paper: "Image-Based malware classification using ensemble of CNN architectures (IMCEC), A novel PCA–whale optimization-based deep neural network model for classification of tomato plant diseases using GPU, Deep learning and medical image processing for coronavirus (COVID-19) pandemic: A survey".

Response: Thank you for your comments. According to your suggestion, we have added references to the discussion, in order to broaden the scope of paper. The added sentences are as follows:

The most common tasks in ML research are regression [56], classification [57-59], clustering [60], and dimensionality reduction [61, 62]. ML algorithms are widely used in research fields including plant diseases [63], disease diagnosis [57], and remote sensing image processing [64-66]. (Line304-306 Page 17 in the revised manuscript)

56. Sharma S, Ahmed S, Naseem M, Alnumay WS, Cho GH. A Survey on Applications of Artificial Intelligence for Pre-Parametric Project Cost and Soil Shear-Strength Estimation in Construction and Geotechnical Engineering. Sensors. 2021;21(2):463. https://doi.org/10.3390/s21020463. PMID: 33440731.

57. Sb A, Pkrm A, Qvp B, Trg A, Srks A, Clc A, et al. Deep learning and medical image processing for coronavirus (COVID-19) pandemic: A survey. Sustainable Cities and Society. 2020. https://doi.org/10.1016/j.scs.2020. 102589. PMID: 33169099.

58. Vasan D, Alazab M, an SWs, Safaei B, Zheng Q. Image-Based malware classification using ensemble of CNN architectures (IMCEC). Computers & Security. 2020;92:101748. https://doi.org/10.1016/j.cose.2020.101748.

59. He Q, Yu S, Xu H, Liu J, Du Y. A Weighted Threshold Secret Sharing Scheme for Remote Sensing Images Based on Chinese Remainder Theorem. Computers, Materials and Continua. 2019;58(2):349-61. https://doi.org/10.32604 /cmc. 2019.03703.

60. Moayedi Y, Somerset E, Fan S, Doumouras B, Teuteberg JJ. Predicting Cardiac Allograft Vasculopathy Profiles Using Machine Learning Clustering. The Journal of Heart and Lung Transplantation. 2021;40(4):S40. https://doi.org/10.1016/j.healun.2021.01.1836.

61. Gadekallu TR, Khare N, Bhattacharya S, Singh S, Alazab M. Early Detection of Diabetic Retinopathy Using PCA-Firefly Based Deep Learning Model. Electronics. 2020;9(2):274. https://doi.org/10.3390/electronics9020274.

62. Gadekallu TR, Khare N, Bhattacharya S, Singh S, Srivastava G. Deep neural networks to predict diabetic retinopathy. Journal of Ambient Intelligence and Humanized Computing. 2020;(13). https://doi.org/10.1007/ s12652-020-01963-7. 

63. Gadekallu TRR, Dharmendra Singh, Reddy M. Praveen Kumar, Lakshmanna Kuruva, Bhattacharya Sweta, Singh Saurabh, et al. A novel PCA-whale optimization-based deep neural network model for classification of tomato plant diseases using GPU. Journal of Real-Time Image Processing. Journal of Real-Time Image Processing. 2020. https://doi.org/10.1007/s11554-020-00987-8 .

64. Corral-Plaza D, Boubeta-Puig J, Ortiz G, Garcia-De-Prado A. An Internet of Things Platform for Air Station Remote Sensing and Smart Monitoring. Computer Systems Science and Engineering. 2020;35(1):5-12. https://doi.org/10.32604/csse.2020.35.005.

65. Liu X, Yu J, Song W, Zhao X, Wang A. Remote Sensing Image Classification Algorithm Based on Texture Feature and Extreme Learning Machine. Computers, Materials and Continua. 2020;65(2):1385-95. https://doi.org/10.32604/ cmc.2020.011308.

66. Xuhan, Yang H, Shen Q, Yang J, Shuang C. Automatic Terrain Debris Recognition Network Based on 3D Remote Sensing Data. Computers, Materials and Continua. 2020;65(1):579-96. https://doi.org/10.32604/cmc.2020. 011262.

4. Justify why the authors choose the ML algorithms used in the paper.

Response: Thank you for your comments. According to your suggestion, we explain the reasons for choosing the ML algorithms in section of introduction. The added sentences are as follows:

These two algorithms are suitable for the resampling and feature selection of high-dimensional data with many features [30, 31]. Moreover, these two algorithms have been applied well in disease diagnosis and image processing [32, 33]. (Line75-77 Page 7 in the revised manuscript)

30. Wright MN, Ziegler A. ranger: A Fast Implementation of Random Forests for High Dimensional Data in C++ and R. Journal of Statal Software. 2017;77(1):1-17. https://doi.org/10.18637/jss.v077.i01. PMID: 20505004.

31. Coelho F, Costa M, Verleysen M, Braga AP. LASSO multi-objective learning algorithm for feature selection. Soft Computing. 2020;24(1-4). https://doi.org/10.1007/s00500-020-04734-w.

32. Patil AR, Kim S. Combination of Ensembles of Regularized Regression Models with Resampling-Based Lasso Feature Selection in High Dimensional Data. Mathematics 2020, 8, 110. https://doi.org/10.3390/math8010110.

33. Luo S, Chen Z. Feature Selection by Canonical Correlation Search in High-Dimensional Multiresponse Models With Complex Group Structures. Journal of the American Statistical Association. 2020;115. https://doi.org/10.1080/01621459.2019.1609972. 

5. Compare the current work with recent state-of-the-art.

Response: Thank you for your comments. According to your suggestion, we add a table (table 1) as “Summary of spectral band selection and modelling techniques for soil properties prediction”. Compare the current work with recent state-of-the-art. The added sentences are as follows:

After reviewing the state-of-the-art literature, it is evident that exploring the most suitable band selection and selecting the appropriate combination of modeling methods are still the key problems in the field of hyperspectral prediction modeling of soil properties. ML techniques have proved to be effective in dealing with large amounts of soil spectral variables [26-29]. The techniques mentioned above have been applied to obtain prediction models of soil properties. However, few studies have mentioned the application of Ranger and least absolute shrinkage and selection operator (Lasso) algorithms to spectral band selection. These two algorithms are suitable for the resampling and feature selection of high-dimensional data with many features [30, 31]. Moreover, these two algorithms have been applied well in disease diagnosis and image processing [32, 33]. One of the goals of this study was to determine whether ML techniques can improve the selection of spectral bands. (Line69-78 Page 7 in the revised manuscript)

26. Zuo R, Xiong Y, Wang J, Carranza EJM. Deep learning and its application in geochemical mapping. Earth-Science Reviews. 2019;02:023. https://doi.org/10.1016/j.earscirev.2019.02.023.

27. Patel AK, Ghosh JK, Pande S, Sayyad S. Deep Learning-Based Approach for Estimation of Fractional Abundance of Nitrogen in Soil from Hyperspectral Data. IEEE Journal of Selected Topics in Applied Earth Observations and Remote Sensing. 2020. https://doi.org/10.1109/JSTARS.2020.3039844.

28. Yang J, Wang X, Wang R, Wang H. Combination of Convolutional Neural Networks and Recurrent Neural Networks for predicting soil properties using Vis–NIR spectroscopy. Geoderma. 2020;380:114616. https://doi.org/10.1016/ j.geoderma.2020.114616.

29. Barra I, Haefele SM, Sakrabani R, Kebede F. Soil spectroscopy with the use of chemometrics, machine learning and pre-processing techniques in soil diagnosis: Recent advances -A review. TrAC Trends in Analytical Chemistry. 2020;135. https://doi.org/10.1016/j.trac.2020.116166.

30. Wright MN, Ziegler A. ranger: A Fast Implementation of Random Forests for High Dimensional Data in C++ and R. Journal of Statal Software. 2017;77(1):1-17. https://doi.org/10.18637/jss.v077.i01. PMID: 20505004.

31. Coelho F, Costa M, Verleysen M, Braga AP. LASSO multi-objective learning algorithm for feature selection. Soft Computing. 2020;24(1-4). https://doi.org/10.1007/s00500-020-04734-w.

32. Patil AR, Kim S. Combination of Ensembles of Regularized Regression Models with Resampling-Based Lasso Feature Selection in High Dimensional Data. Mathematics 2020, 8, 110. https://doi.org/10.3390/math8010110.

33. Luo S, Chen Z. Feature Selection by Canonical Correlation Search in High-Dimensional Multiresponse Models With Complex Group Structures. Journal of the American Statistical Association. 2020;115. https://doi.org/10.1080/01621459.2019.1609972. 

6. Discuss about the limitations and future scope of the current work. 

Response: Thank you for your comments. According to your suggestion, we discuss the limitations and future scope of the current work in section of conclusion. The added sentences are as follows:

Considering that soil samples from different soil types and land use patterns usually exhibit significant spectral differences, it may not be feasible to extrapolate our results to other study areas. As future work, the proposed methods could be implemented on hyperspectral image band selection and modeling. This technique is expected to provide technical support for the non-destructive and large-scale monitoring of forest soil health assessment. (Line363-368 Page 20 in the revised manuscript)

---

## [Decision Letter · Decision Letter 1]

4 Jun 2021

Hyperspectral Band Selection and Modeling of Soil Organic Matter Content in a Forest Using the Ranger Algorithm

PONE-D-21-09521R1

Dear Dr. Tang,

We’re pleased to inform you that your manuscript has been judged scientifically suitable for publication and will be formally accepted for publication once it meets all outstanding technical requirements.

Kind regards,

Thippa Reddy Gadekallu

Academic Editor

PLOS ONE

Additional Editor Comments (optional):

Reviewers' comments:

Reviewer's Responses to Questions

**Comments to the Author**

1. If the authors have adequately addressed your comments raised in a previous round of review and you feel that this manuscript is now acceptable for publication, you may indicate that here to bypass the “Comments to the Author” section, enter your conflict of interest statement in the “Confidential to Editor” section, and submit your "Accept" recommendation.

Reviewer #1: All comments have been addressed

Reviewer #2: All comments have been addressed

2. Is the manuscript technically sound, and do the data support the conclusions?

Reviewer #1: Yes

Reviewer #2: Yes

3. Has the statistical analysis been performed appropriately and rigorously? 

Reviewer #1: Yes

Reviewer #2: Yes

4. Have the authors made all data underlying the findings in their manuscript fully available?

Reviewer #1: Yes

Reviewer #2: Yes

5. Is the manuscript presented in an intelligible fashion and written in standard English?

Reviewer #1: Yes

Reviewer #2: Yes

6. Review Comments to the Author

Reviewer #1: the paper "Hyperspectral Band Selection and Modeling of Soil Organic Matter Content in a Forest Using the Ranger Algorithm" proposed by authors have done all comments addressed. the paper can go for publication.

Reviewer #2: The authors have done a good job in addressing all the comments and suggestions. The paper is improved significantly and is in a good shape now. I recommend the paper to be accepted in the current form.

7. PLOS authors have the option to publish the peer review history of their article (what does this mean?). If published, this will include your full peer review and any attached files.

Reviewer #1: No

Reviewer #2: No

---

## [Editor Report · Acceptance letter]

17 Jun 2021

PONE-D-21-09521R1 

Hyperspectral Band Selection and Modeling of Soil Organic Matter Content in a Forest Using the Ranger Algorithm 

Dear Dr. Tang:

I'm pleased to inform you that your manuscript has been deemed suitable for publication in PLOS ONE. Congratulations! Your manuscript is now with our production department. 

Kind regards, 

on behalf of

Dr. Thippa Reddy Gadekallu 

Academic Editor

PLOS ONE